# Comparison of Surface Water Volume Estimation Methodologies That Couple Surface Reflectance Data and Digital Terrain Models

**Ignacio Fuentes *** , **José Padarian, Floris van Ogtrop** and **R. Willem Vervoort**

School of Life and Environmental Sciences, The University of Sydney, New South Wales 2006, Australia; jose.padarian@sydney.edu.au (J.P.); floris.vanogtrop@sydney.edu.au (F.v.O.); willem.vervoort@sydney.edu.au (R.W.V.)
* Correspondence: ignacio.fuentes@sydney.edu.au

**Abstract:** Uncertainty about global change requires alternatives to quantify the availability of water resources and their dynamics. A methodology based on different satellite imagery and surface elevation models to estimate surface water volumes would be useful to monitor flood events and reservoir storages. In this study, reservoirs with associated digital terrain models (DTM) and continuously monitored volumes were selected. The inundated extent was based on a supervised classification using surface reflectance in Landsat 5 images. To estimate associated water volumes, the DTMs were sampled at the perimeter of inundated areas and an inverse distance weighting interpolation was used to populate the water elevation inside the flooded polygons. The developed methodology (IDW) was compared against different published methodologies to estimate water volumes from digital elevation models, which assume either a flat water surface using the maximum elevation of inundated areas (Max), and a flat water surface using the median elevation of the perimeter of inundated areas (Median), or a tilted surface, where water elevations are based on an iterative focal maximum statistic with increasing window sizes (FwDET), and finally a tilted water surface obtained by replacing the focal maximum statistic from the FwDET methodology with a focal mean statistic (FwDET_mean). Volume estimates depend strongly on both water detection and the terrain model. The Max and the FwDET methodologies are highly affected by the water detection step, and the FwDET_mean methodology leads to lower volume estimates due to the iterative smoothing of elevations, which also tends to be computationally expensive for big areas. The Median and IDW methodologies outperform the rest of the methods, and IDW can be used for both reservoir and flood volume monitoring. Different sources of error can be observed, being systematic errors associated with the DTM acquisition time and the reported volumes, which for example fail to consider dynamic sedimentation processes taking place in reservoirs. Resolution effects account for a fraction of errors, being mainly caused by terrain curvature.

**Keywords:** flood mapping; water volumes; remote sensing; Google Earth Engine

## 1. Introduction

Changes in global climate and population result in increased uncertainty in relation to production and resource exploitation [1–3]. This is particularly relevant for water resources, whose availability and projections have recently been disputed [4,5]. Given this uncertainty, alternatives to quantify the availability of water resources must be developed to define water management plans or risk assessments with higher accuracy [6,7].

Remote sensing is an important tool for studying surface water [8–10]. It has the advantage that it can be applied in conjunction with other direct measurements, and provides not only snapshots



of ongoing processes, but can also capture the temporal fluctuation and seasonality of surface water processes [11,12].

The advantages of remote sensing data compared to other hydrological data lie in the opportunity to account for the spatial variability of processes [8,13]. Thus, while gauge and meteorological stations provide data for a specific location, satellite imagery reflects a larger spatial area [14]. Satellites also regularly pass over the same location, which provides a time series of the images, catching the temporal variability of some processes [7,15].

Most examples of water detection from space focus on the spatial extent of surface water, but not necessarily on the volume quantification [15,16]. In the cases where quantity is studied, they are generally coupled with gauge station measurements, bathymetry, or digital terrain models (DTM), to estimate water volumes [7,11,17,18]. Several methods have been developed to obtain the depth of surface water. However, the performance of these methods has been mainly assessed against field measurements through individual events, rather than against the temporal dynamics of surface hydrological processes [17]. One example where temporal changes in hydrological processes were been taken into account only covered short time periods [7].

Flood studies using remote sensing have mostly been limited to short periods or single events due to challenges in acquiring detailed remote sensing information, which requires high computational storage capacity due to the efforts involved in pre-processing imagery [19]. However, several alternatives have recently been developed to cope with these tasks. One of the most important has been the development of the Google Earth Engine platform, which has multi-petabyte processed and regularly updated geospatial datasets as well as a wide range of algorithms that facilitate spatial analysis and remote sensing functionality [20,21].

A second main difficulty is to verify predicted inundation volumes, and this is because of the lack of a frame of reference [22]. Generally, the solution is to use water levels from gauging stations and dams or the estimation of the components of the water budget, which usually provide a rough estimate of the overall water availability [23], but do not necessarily provide space and time verification. In addition, the scarcity of bathymetric continental data means that estimating surface water volumes in permanent water bodies is difficult [13]. Moreover, irregularly inundated areas, in which water was absent when the DTMs were derived, are not consistently surveyed.

While the most common methodology for estimating surface water volumes assumes a flat water surface [7], this is rarely the case. In flood processes for instance, the flow will be influenced by the topography of the terrain over which the water is passing. Even in big reservoirs and lakes, water surfaces are not totally flat. This may be caused by seiches or drain exits [24]. Therefore, assuming flat water surfaces for water volume estimations may lead to substantial errors in water availability. The main objective of this study was to assess an automated methodology to estimate surface water volumes for flood events and reservoirs, taking into account that inundated areas are not necessarily flat. We then compared this methodology with previously developed alternatives, which use surface reflectance imagery and elevation models as inputs. The final aim is to improve the calculation of a flood time series using remote sensing data.

## 2. Materials and Methods

### 2.1. Study Areas and Data

The study was carried out at nine locations that have digital terrain models (DTM) and where water volumes are continuously monitored. The first case study was the Menindee Lakes (Cawndilla, Menindee, and Pamamaroo), which are located in NSW, Australia (Figure 1). These lakes were modified in 1968 in order to increase the storage capacity and control floods in the Murray Darling basin [25]. A hot and dry climate on the floodplain depressions of the Lower Darling River characterizes the region where these lakes are located [26]. These characteristics in combination with regular floods lead to water

fluctuations in the lakes, suitable for this analysis. Some characteristics of the Menindee lakes are shown in Table 1.

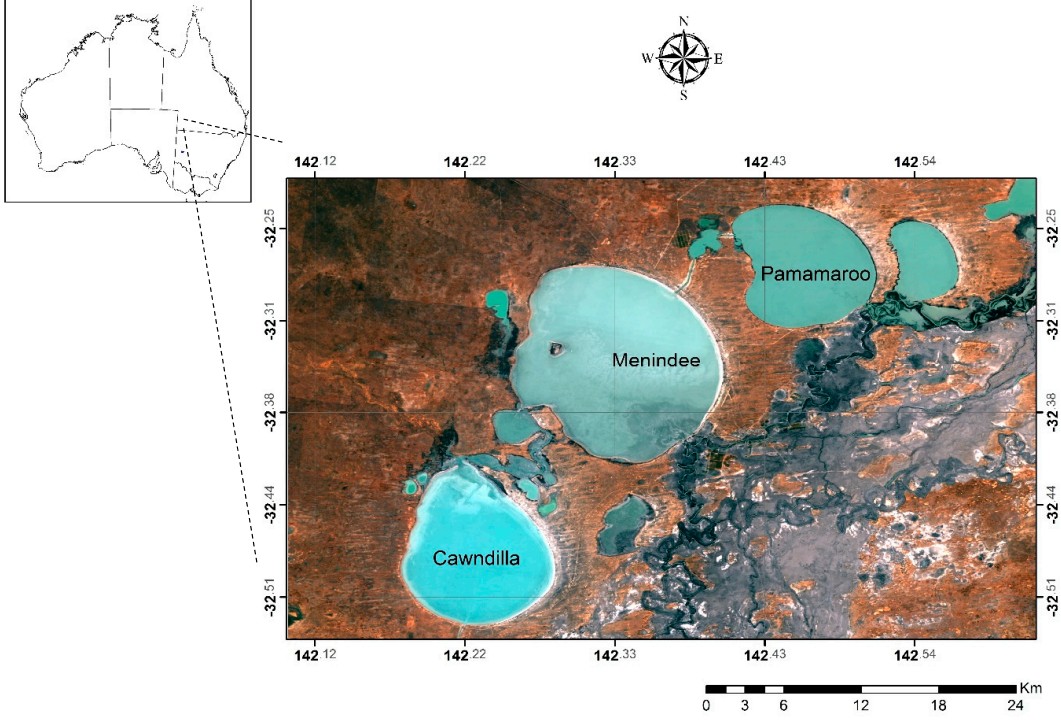

**Figure 1.** Location of Menindee lakes in NSW, Australia.

Other surface water bodies assessed were the Atoka reservoir, and the Ellsworth, Fork, Ray Roberts, Hubbard Creek, Tawakoni, and Stanley Draper lakes, located in Oklahoma and Texas in the United States (Figures 2 and 3). The Oklahoma reservoirs are surrounded by smooth topographies within the Mississippi river basin and characterized by a humid subtropical climate [27].

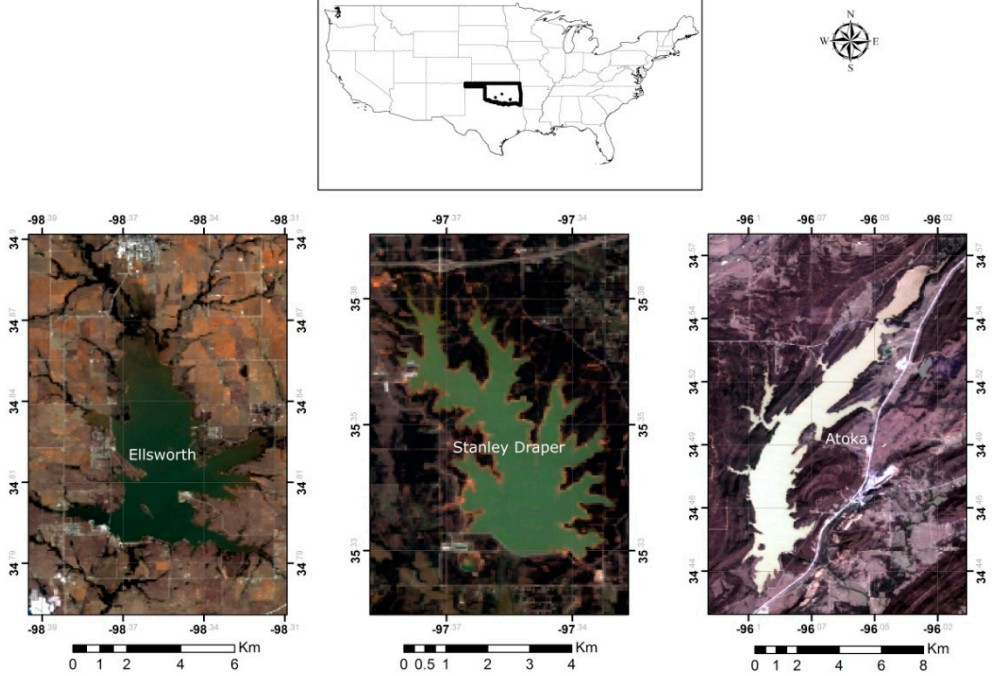

**Figure 2.** Location of Oklahoma water reservoirs in the United States.

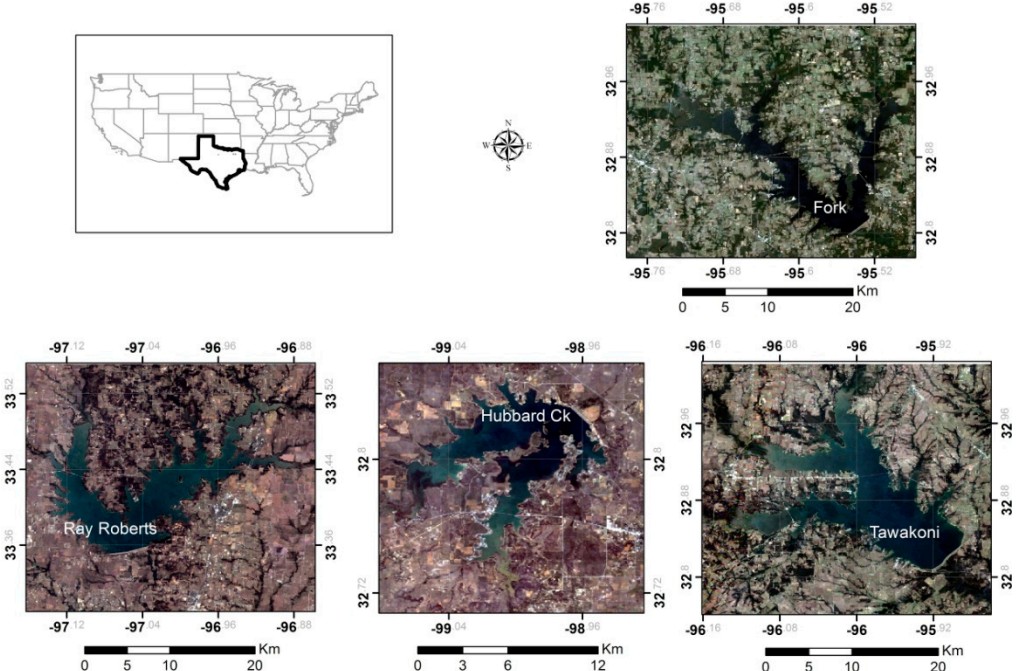

**Figure 3.** Texas water reservoirs selected.

The Texas reservoirs were selected based on the available information provided by the Texas Water Development Board webpage (https://www.twdb.texas.gov/). These lakes are located in northeast Texas, and also have a subhumid tropical climate (Figure 3). The primary purpose of these reservoirs is for water supply and conservation, but flood control is also an important purpose, at least for the Hubbard Creek and Ray Roberts lakes.

In the case of the Menindee lakes, the reference elevation data corresponds to the available Light Detection And Ranging (LiDAR) DTM at a 5 m resolution [28]. The campaign that obtained the LiDAR elevations covering the Menindee lakes was carried out in 2009, during a period in which all three reservoirs were empty, and therefore the observed elevation corresponds to the elevation of the bottom of the lakes (Figure 4).

For the Oklahoma lakes, bathymetric maps were obtained from the Oklahoma Water Resources Board with a resolution of 1.5 m (https://www.owrb.ok.gov/), which were resampled at 3 m and subsequently superimposed on the USGS National Elevation Dataset, which presents a $^1/_3$ arc-second resolution [29] (Figure 5). However, the bathymetric maps were referenced to the specific normal elevation of each reservoir, instead of being referenced to the water elevation at which they were surveyed.

In the case of Texas reservoirs, the bathymetric maps were obtained from the Texas Water Development Board webpage. The bathymetric studies were carried out in different surveys between 2008 and 2018 using multi-frequency sub-bottom profiling depth sounders [30–33]. From elevation contour lines, bathymetric images were obtained at a 3 m resolution and superimposed on the USGS National Elevation Dataset (Figure 6).

Landsat 5 surface reflectance imagery was used to detect the surface water. The images were masked to remove clouds and cloud shadows. Images were used for the entire period of the operation of the satellite, but a filter was applied such that masked images which contained less than 99% of the reservoir extent were removed.

The recurrence layer of the Global Surface Water (GSW) Mapping Layers, v.1.0 from the Joint Research Centre [34] was used as an input for the water detection using Landsat, available as a dataset in the Google Earth Engine platform.

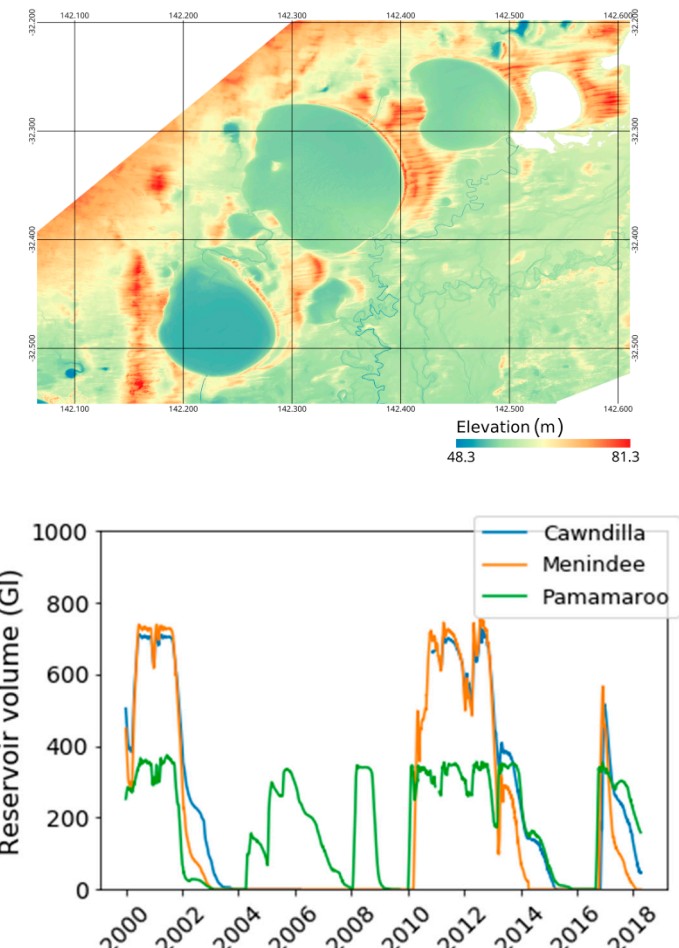

**Figure 4.** Time series of volume storages for the Menindee lakes and LiDAR elevations. One GL is equivalent to $10^6$ m$^3$.

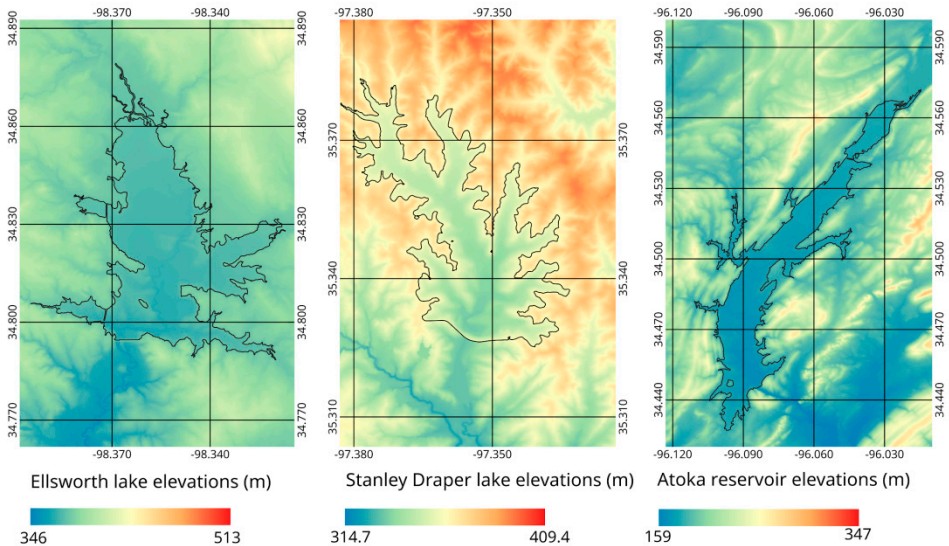

**Figure 5.** Terrain elevation of Oklahoma lakes.

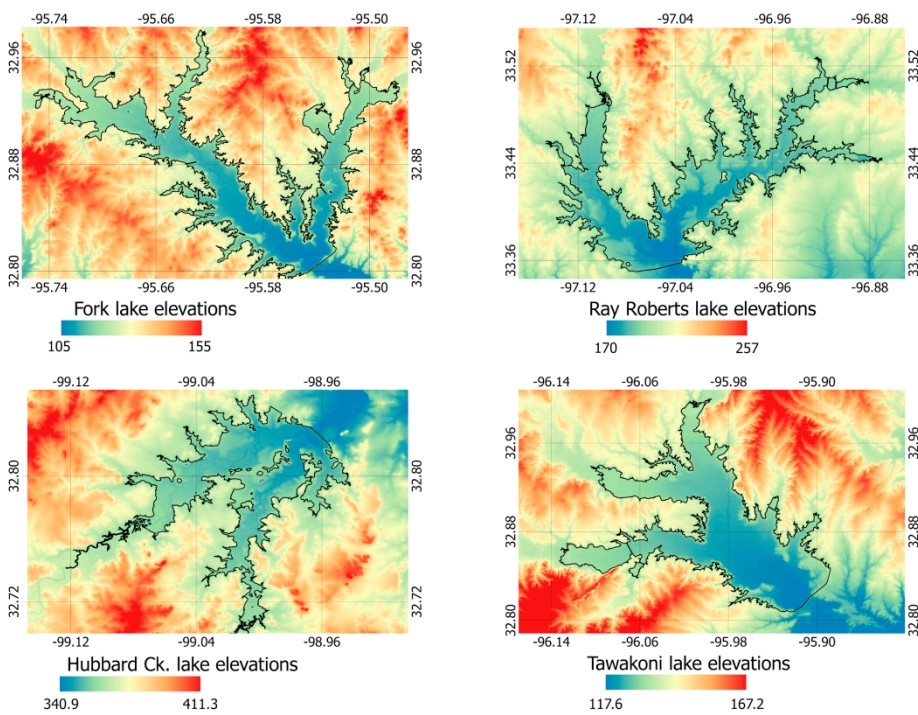

**Figure 6.** Terrain elevation of Texas lakes.

**Table 1.** Design characteristics of the studied reservoirs [30–33,35].

| Reservoir | Storage Capacity (GL) | Area (m$^2$) | Maximum Depth (m) |
|---|---|---|---|
| Cawndilla lake | 631.05 | 94,851,864 | 8.7 |
| Menindee lake | 629.49 | 163,936,661 | 8.1 |
| Pamamaroo lake | 277.73 | 66,861,857 | 7.8 |
| Ellsworth lake | 100.60 | 20,691,580 | 16.5 |
| Stanley Draper lake | 183.00 | 12,000,000 | 30.0 |
| Atoka dam | 152.00 | 23,000,000 | 18.3 |
| Fork lake | 785.11 | 108,816,018 | 18.29 |
| Ray Roberts lake | 972.59 | 115,926,351 | 32.31 |
| Hubbard Creek lake | 392.46 | 63,483,092 | 18.29 |
| Tawakoni lake | 1075.22 | 151,049,049 | 19.23 |

## 2.2. Water Detection

Munasinghe et al. [36] argue that the best performance for water detection methods on Landsat imagery is obtained using supervised classification, rather than using normalized indices. Therefore, a classification and regression tree analysis (CART) was applied to the reflectance bands of the Landsat imagery to delineate inundated areas by selecting known surface water and dry land end-members. This was done by drawing polygons classified as water on several images, previously masked to remove clouds and cloud shadows. Other polygons were also delineated on dry areas with different land covers, and classified as dry polygons, which produced surface water classified images (Figure 7).

A total of 188 and 205 polygons in surface water and dry land areas were used to train the classifier, respectively, using 15 different Landsat 5 images covering Australia and the United States on different dates. Nevertheless, since the classifier is fed by pixels rather than polygons, the classification was carried out using 56,438 dry land and 133,194 surface water pixels.

Due to the difficulties caused by topography and dark lithologies, in which mountain shadows and dark lithologies tend to be classified as water due to the low reflectance, the recurrence layer of the GSW dataset [34] was also used as an input for the classification. This was appended as a band to the surface reflectance bands of the Landsat images and passed to the CART classifier.

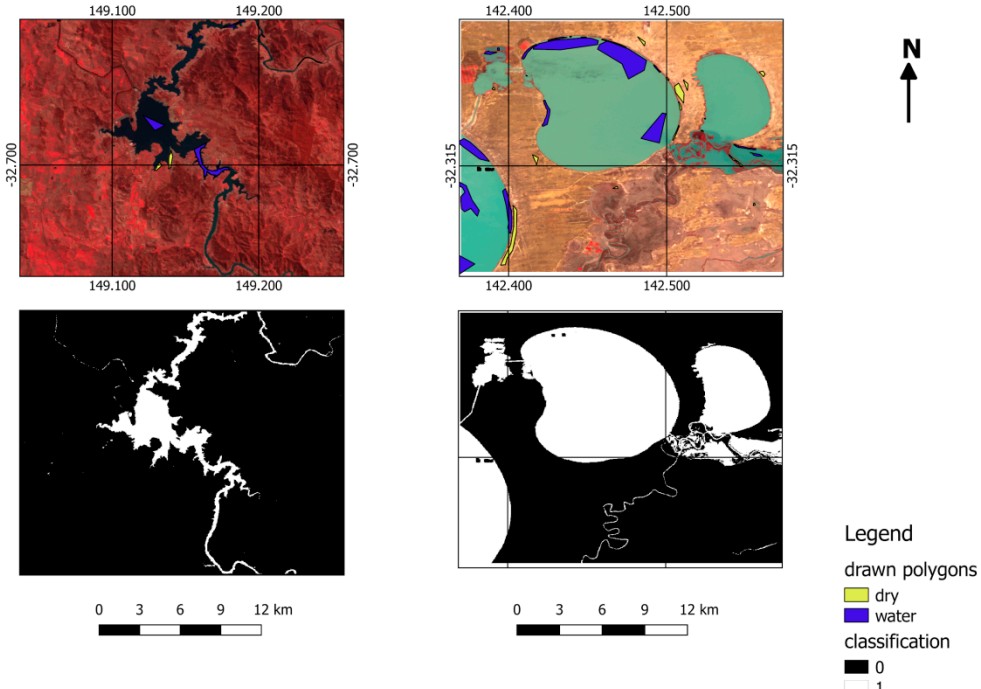

**Figure 7.** Two examples of areas where polygons classified as water (blue polygons) and dry (yellow polygons) were drawn to feed the classification and regression tree analysis (CART) classifier with their corresponding classifications. The background images where the polygons were drawn correspond to color infrared Landsat images.

## 2.3. Water Depths

Once the inundated areas in the images were delineated, different methods were used to estimate the water volumes. This involves using the DTM products, including bathymetries, to obtain the depth of water at the surface, which was subsequently multiplied by the area of inundated pixels.

The first method (Max) assumes that the water surface during floods is flat, based on Siev et al. [7] who studied the seasonal change in water volumes in a floodplain of almost 5000 km$^2$ in Cambodia. In it, several polygons with inundated areas are overlain by a DTM, and the maximum elevation of water in those polygons is assumed to be the elevation of the surface water. Subsequently, the DTM is subtracted from these elevations in each polygon to get the water depth.

Since the Max methodology may be strongly affected by errors in the surface water classification, it was complemented by another hydro-flattening methodology, subsequently referred to as "Median". It consisted of a line vectorization of the perimeter of inundated areas, which was then buffered 2.5 m at each side. Subsequently, the DTM was clipped by the buffered layer extent and the median contour DTM elevation was estimated and extrapolated for each inundated area. Finally, the DTM was subtracted from the median elevation within the perimeter of inundated areas to get the water depths.

The third method (FwDET) was developed by Cohen et al. [17] for flood analysis. It involves the conversion of inundated areas into polygons to obtain the elevations at the perimeter of polygons. Subsequently, it applies a focal statistic (focal maximum) in a series of iterations with increasing window sizes to populate the area inside the polygons with water elevations. The final stage involves subtracting the water elevations from the original DTM to get the water depths. Negative water depths are converted to 0, and a final low-pass filter with a kernel of 3 pixels is used to smooth any abrupt change in the water elevations. An important detail is that the number of iterations corresponds to the minimum number of iterations needed to completely populate all the inundation polygons. Additionally, a modification of the FwDET methodology was implemented (subsequently referred to as FwDET_mean), which replaced the focal maximum statistic of the original methodology with a focal mean statistic.

The last method, inverse distance weighting (IDW), also corresponds to a modification of the FwDET algorithm to improve volume estimates in water reservoirs, because the Cohen et al. [17] study reports methodological errors in the estimation of reservoir volumes. The new methodology consists of delineating the perimeter of inundated areas and applies a random sampling of the perimeter elevations using a buffer of 2.5 m on each side of the perimeter contour, setting the number of sampling points to 5000. Then, an inverse distance weighting interpolation is applied to the sampled points to obtain the elevation of the water, which is subsequently subtracted from the DTM to obtain the water depth (Figure 8).

A filter was applied to all methods, such that polygons with less than six inundated pixels were removed from the Landsat images.

All preprocessing steps, the water detection, and the different methodologies were implemented in Google Earth Engine.

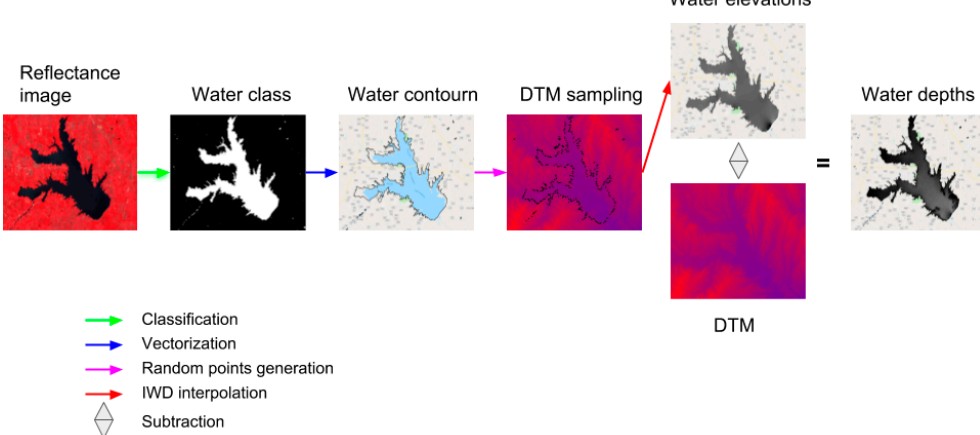

**Figure 8.** Schema of inverse distance weighting interpolation (IDW) methodology for water volume estimation.

### 2.4. Covariates and Performance of Methodologies

In the case of Cawndilla lake, an additional 1 m resolution LiDAR DTM was also obtained to assess how the resolution of the DTM affects the volume estimations. Thus, for the Cawndilla and the Hubbard lakes (which already have 1 m DTMs), the original resolution was progressively reduced to 3, 5, 10, 20, 30, 40, 50, 100, 200, 300, 400, 500, 1000, 2000, and 3000 m. The analysis of the slope and its effect on the estimates compared the slope of both reservoirs and the performance of the volume estimates at the different resolutions.

The final volume estimates were compared against volumes reported by government organizations from gauging station data. In the case of the Menindee lakes, reported daily volumes were obtained from the Water NSW webpage (https://realtimedata.waternsw.com.au/), whilst daily volumes from the Oklahoma and Texas reservoirs were obtained from the USGS platform (https://waterdata.usgs.gov/nwis). In the case of the USGS datasets, only volumes that were approved for publication were used. Reported volumes are obtained based on ratings tables generated from initial bathymetric surveys, which provide the relationship between storage levels and dam volumes.

As a performance evaluation the root mean square error and the bias of the relationship between observed and estimated volumes were estimated, in addition to the coefficient of determination and a *p*-value for the linear regression between both datasets.

## 3. Results

### 3.1. Menindee Lakes

As the LiDAR DTM used in the volume estimates of the Menindee lakes was developed when no water was stored, quantifying water stored in the reservoirs does not require further processing of the

terrain model. In addition, the DTM can be extrapolated to quantify water volumes associated with flood events in areas that are dry most of the time. A summary of the performance of the different methodologies for the Menindee lakes is presented in Table 2.

**Table 2.** Performance of the different methodologies used to estimate water volumes in the Menindee lakes.

| Reservoir | Method | *p*-Value | $R^2$ | RMSE (GL or $m^3 \times 10^6$) | Bias (GL or $m^3 \times 10^6$) |
|---|---|---|---|---|---|
| Cawndilla | Max | 0.0 | 0.96 | 88.25 | 40.65 |
| | Median | 0.0 | 0.99 | 85.85 | −53.91 |
| | FwDET | 0.0 | 0.98 | 56.27 | 21.75 |
| | FwDET_mean | 0.0 | 0.98 | 174.69 | −111.59 |
| | IDW | 0.0 | 0.99 | 81.86 | −49.32 |
| Menindee | Max | 0.0 | 0.95 | 254.59 | 143.30 |
| | Median | 0.0 | 0.99 | 60.96 | −36.92 |
| | FwDET | 0.0 | 0.97 | 250.22 | 129.17 |
| | FwDET_mean | 0.0 | 0.98 | 164.03 | −105.79 |
| | IDW | 0.0 | 0.99 | 59.24 | −36.74 |
| Pamamaroo | Max | 0.0 | 0.80 | 141.79 | 107.47 |
| | Median | 0.0 | 0.97 | 48.62 | −41.28 |
| | FwDET | 0.0 | 0.85 | 113.63 | 93.63 |
| | FwDET_mean | 0.0 | 0.89 | 105.31 | −94.98 |
| | IDW | 0.0 | 0.96 | 48.72 | −41.18 |

Except for the Cawndilla lake, the Max method tends to have higher noise compared to the other methods and tends to overestimate the volumes. This methodology appears more sensitive to the water detection technique. While other methodologies use the entire perimeter of the surface water, which offsets errors generated in the delineation of surface water, the maximum elevation methodology, by picking up only one value of elevation and extrapolating this to the entire water surface, is prone to errors at the water detection step.

The FwDET methodology leads to high $R^2$ coefficients, but also to high errors in the Menindee and Pamamaroo lakes. In this case, the bias is positive due to a propagation of the maximum elevations in the successive iterations of the focal statistic, and therefore propagates the errors caused in the water detection and by the mismatch between the resolutions of the DTM and the surface reflectance product. The FwDET_mean, the Median, and the inverse distance weighting interpolation estimates result in a negative bias, which was higher for the first method (Table 2). When using the FwDET_mean method, increasing the iterations, which increased the window size in the focal mean statistic, tends to smooth and diminish the volumes. Therefore, the minimum number of iterations was based on the number of iterations needed to fill the entire area inside the inundated polygons [17]. However, despite this, considerable negative bias is introduced. Another disadvantage of this methodology is that the focal statistic iteration with increasing window sizes is computationally expensive for big areas at high resolutions, which constrains its use.

Using both the Median and the inverse distance weighting interpolation methodology decreases the errors and the bias, improving the volume estimates. By assessing the residuals between estimated and observed values using the IDW method it is clear that the negative bias increases at higher observed volumes (Figure 9b).

From the analysis, a relationship between inundated areas and volumes can be derived (Figure 9c), which facilitates further analysis of lake volume by just fitting a regression curve to the data. The shape of the curves also identifies characteristics of the reservoirs. In this case, the Menindee lakes have an exponential increase in volumes with respect to the inundated area, which implies small slopes inside the lakes, but a sharp slope at the perimeter of the lakes, which can also be inferred from the circular shape of the lakes in Figure 4.

The area/volume relationship indicates a steeper slope for greater inundated areas. The slope is lower at the Cawndilla lake and is confirmed by the cumulative frequency of slopes from the lakes (Figure 9d).

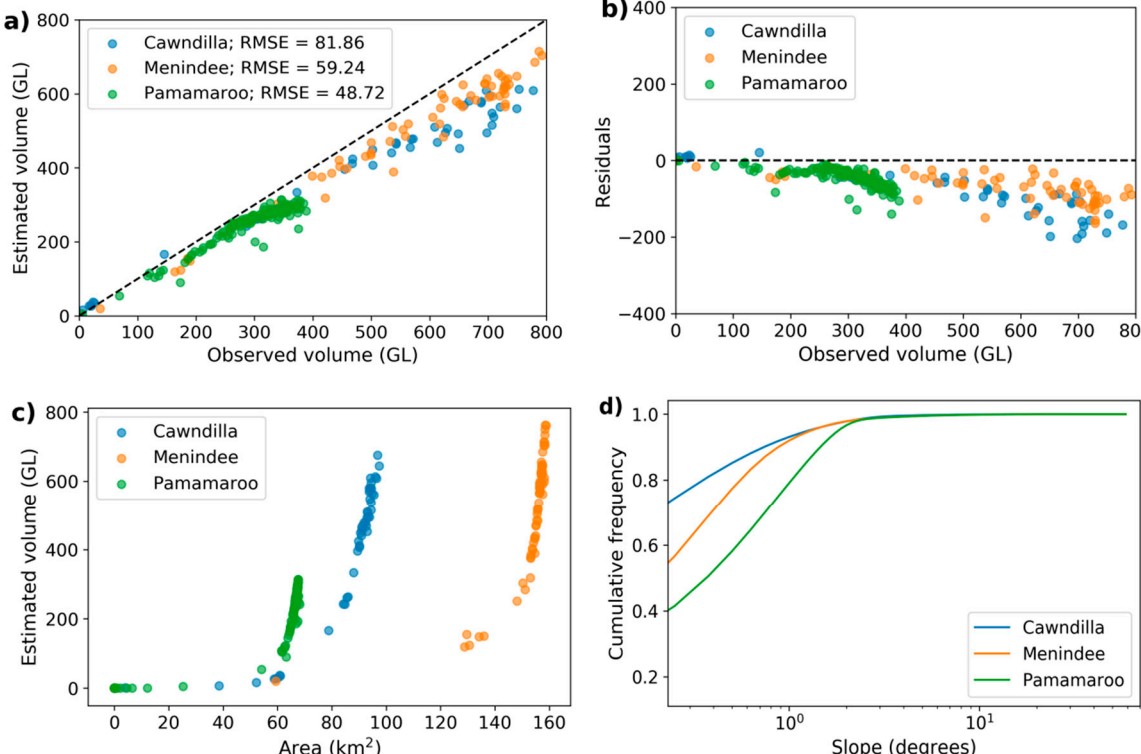

**Figure 9.** Water volume estimates (**a**), residuals (**b**) and area–volume relationship (**c**) obtained using the inverse distance weighting interpolation methodology on the Cawndilla, Menindee, and Pamamaroo lakes. The cumulative frequency of slopes in the reservoirs is also presented (**d**).

### 3.2. Oklahoma Reservoirs

The Oklahoma lakes have significantly smaller storage volumes, and as a result have generally smaller root mean square errors (RMSEs) than the Menindee lakes, despite their lower determination coefficients (Table 3). However, all reservoirs have a systematic bias, which was always negative for the FwDET_mean and positive for the Max and FwDET methodologies over the entire range of observations. In the case of the Median and IDW methods, the bias is both negative and positive, and as in the Menindee lakes, they have the lower RMSE values.

**Table 3.** Performance of the different methodologies used to estimate water volumes in Oklahoma reservoirs.

| Reservoir | Method | $p$-Value | $R^2$ | RMSE (GL or $m^3 \times 10^6$) | Bias (GL or $m^3 \times 10^6$) |
|---|---|---|---|---|---|
| Atoka dam | Max | 0.0 | 0.25 | 150.01 | 133.05 |
| | Median | 0.0 | 0.93 | 13.83 | −10.99 |
| | FwDET | 0.0 | 0.28 | 98.03 | 68.03 |
| | FwDET_mean | 0.0 | 0.77 | 68.61 | −67.12 |
| | IDW | 0.0 | 0.77 | 24.79 | −21.81 |
| Ellsworth lake | Max | 0.0 | 0.79 | 68.85 | 67.63 |
| | Median | 0.0 | 0.95 | 11.07 | 9.99 |
| | FwDET | 0.0 | 0.81 | 40.77 | 39.12 |
| | FwDET_mean | 0.0 | 0.95 | 14.46 | −12.13 |
| | IDW | 0.0 | 0.94 | 13.30 | 12.33 |
| Stanley Draper lake | Max | 0.0 | 0.84 | 28.11 | 26.69 |
| | Median | 0.0 | 0.96 | 5.24 | −4.36 |
| | FwDET | 0.0 | 0.79 | 20.46 | 17.85 |
| | FwDET_mean | 0.0 | 0.97 | 24.16 | −24.04 |
| | IDW | 0.0 | 0.96 | 13.17 | −12.87 |

The bias can be explained by the fact that the different bathymetric maps were referenced to the mean water level elevation of the reservoir. This choice is generally made because the surveys usually take several days to complete, and can be finished even in different seasons. However, as the reference elevation does not correspond to the reservoir level at the moment of the survey, it introduces consistent errors throughout the entire range of observations. One solution to this problem might be to use the mean elevation of the surveyed days rather than the mean elevation of the reservoir for the generation of the bathymetry maps, especially if the variation of water depths during the survey is lower than the variation in the entire reservoir monitoring period.

Additionally, this bias might be simply removed from the estimates assuming that it is caused by using the mean level elevation of the reservoir instead of the water level at the moment of the survey. The residuals in the Atoka reservoir showed an abrupt drop at the higher end of the volume observations (Figure 10b). This change is related to the change of DTM from the 3 m resolution bathymetry map to the $^1/_3$ arc-second resolution USGS National Elevation Dataset.

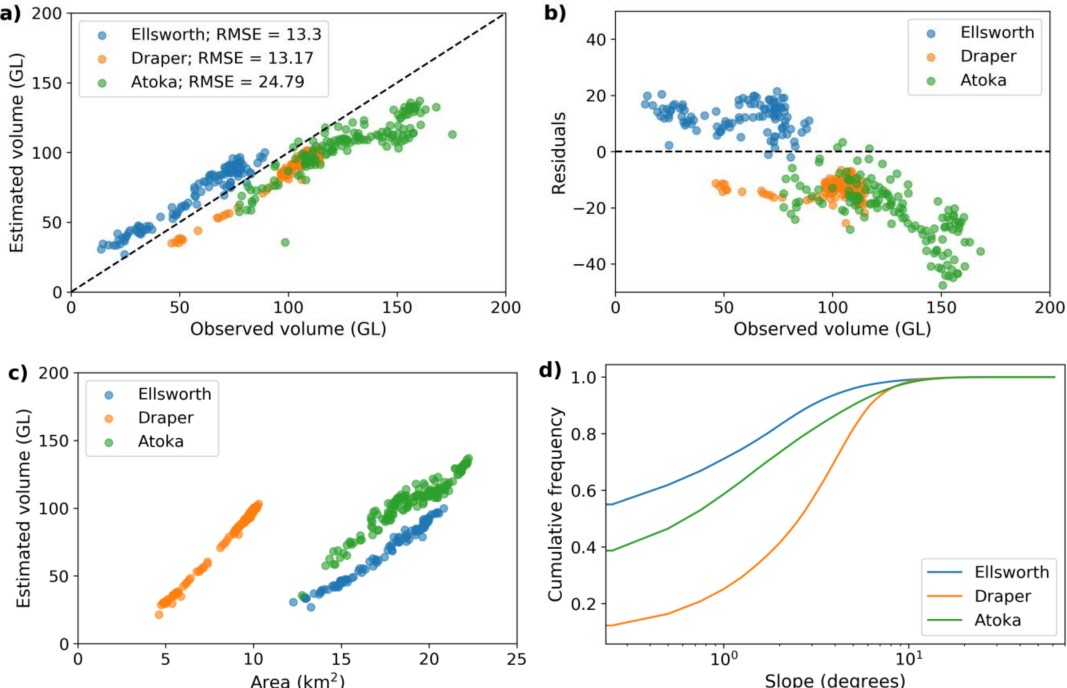

**Figure 10.** Water volume estimations (**a**), their residuals (**b**), and the area–volume relationship (**c**) obtained using the inverse distance weighting interpolation methodology on the Oklahoma reservoirs. The cumulative frequency of slopes in the reservoirs is also presented (**d**).

Analyzing the area–volume relationship (Figure 10c) suggests quite a different behavior compared to the Menindee lakes, as the relationship is almost linear. In this case, the shape of the reservoirs also differs from the circular shape observed in the Menindee lakes, and has less steep slopes at the edges. The steepest slopes can be observed in the Stanley Draper reservoir and the smoothest bathymetry in the Ellsworth lake (Figure 10d).

### 3.3. Texas Reservoirs

The performance of the different methods for the Texas reservoirs is in Table 4. All relationships are highly correlated with the observed data. In general, both the Max and the FwDET methodologies had a positive bias compared to the others.

Again, the Median and IDW methodologies outperform the rest of the methodologies. Comparing the results with the other lakes, it can be observed that there are also increased negative errors for greater storage volumes (Figure 11b). A greater reservoir storage volume relates to a greater flooded

area. Since the perimeter of reservoirs in surface reflectance images is often composed of mixed land surfaces (flooded and dry land), these areas are more susceptible to water detection classification errors. These errors and the mismatch between the resolution of DTMs and the Landsat images can cause errors in the elevations at the perimeter of the reservoirs used to fill the water elevations. This results in bigger errors of volume estimates associated with greater inundated areas.

**Table 4.** Performance of the different methodologies used to estimate water volumes in Texas reservoirs.

| Reservoir | Method | *p*-Value | $R^2$ | RMSE (GL or $m^3 \times 10^6$) | Bias (GL or $m^3 \times 10^6$) |
|---|---|---|---|---|---|
| Hubbard Creek lake | Max | $8.10 \times 10^{-15}$ | 0.90 | 86.71 | 74.68 |
| | Median | $1.90 \times 10^{-23}$ | 0.98 | 22.22 | −12.12 |
| | FwDET | $7.80 \times 10^{-20}$ | 0.96 | 77.15 | 66.78 |
| | FwDET_mean | $6.14 \times 10^{-21}$ | 0.96 | 38.89 | −33.56 |
| | IDW | $5.24 \times 10^{-22}$ | 0.97 | 27.14 | −19.02 |
| Tawakoni lake | Max | $4.19 \times 10^{-15}$ | 0.63 | 288.35 | 262.16 |
| | Median | $8.34 \times 10^{-25}$ | 0.82 | 85.49 | −63.75 |
| | FwDET | $1.47 \times 10^{-23}$ | 0.80 | 263.63 | 219.52 |
| | FwDET_mean | $2.43 \times 10^{-31}$ | 0.89 | 175.90 | −166.11 |
| | IDW | $4.21 \times 10^{-30}$ | 0.88 | 119.07 | −108.63 |
| Ray Roberts lake | Max | $2.51 \times 10^{-47}$ | 0.90 | 188.12 | 162.02 |
| | Median | $4.26 \times 10^{-49}$ | 0.91 | 148.14 | −90.13 |
| | FwDET | $3.98 \times 10^{-47}$ | 0.90 | 172.35 | 145.66 |
| | FwDET_mean | $1.59 \times 10^{-42}$ | 0.87 | 211.32 | −166.59 |
| | IDW | $1.15 \times 10^{-54}$ | 0.93 | 166.04 | −121.30 |
| Fork lake | Max | $1.79 \times 10^{-04}$ | 0.18 | 140.44 | 183.77 |
| | Median | $1.39 \times 10^{-11}$ | 0.48 | 61.29 | −50.81 |
| | FwDET | $1.03 \times 10^{-12}$ | 0.51 | 125.51 | 116.94 |
| | FwDET_mean | $2.37 \times 10^{-14}$ | 0.57 | 126.39 | −122.41 |
| | IDW | $3.10 \times 10^{-11}$ | 0.47 | 81.78 | −74.00 |

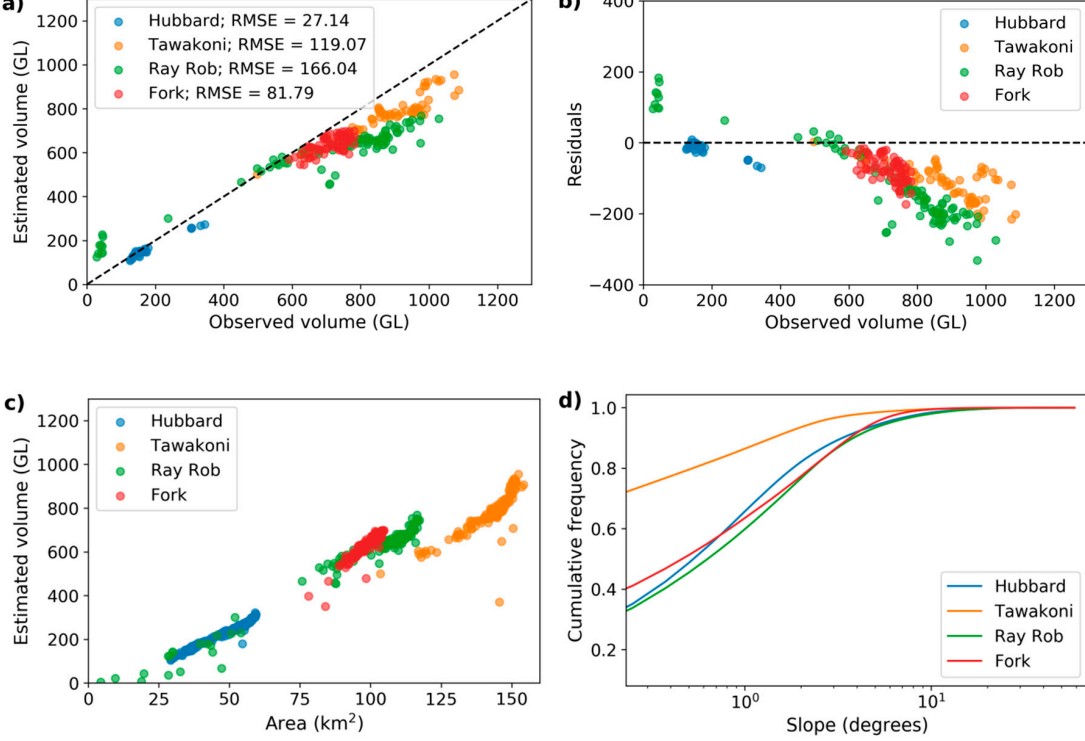

**Figure 11.** Water volume estimations (**a**), their residuals (**b**), and the area–volume relationship (**c**) obtained using the inverse distance weighting interpolation methodology on the Texas reservoirs. The cumulative frequency of slopes in the reservoirs are also presented (**d**).

The relationship between inundated areas and reservoir volumes is again fairly good. In this case the smoothest bathymetry is in the Tawakoni lake (Figure 11d) despite having the highest storage capacity.

A comparison of water depths for the Hubbard Creek lake is shown in Figure 12 for 15 February 1991. It can be observed that there is a clear difference between the Max and the FwDET methodologies and the rest. The Max and the FwDET lead to greater water depths, which may be the result of a combination of causes: mixed land covers within surface reflectance image pixels, a mismatch between resolutions of the DTM and the surface reflectance product, and errors in the water detection step. The best performers, the Median and IDW methodologies, give similar water depth maps.

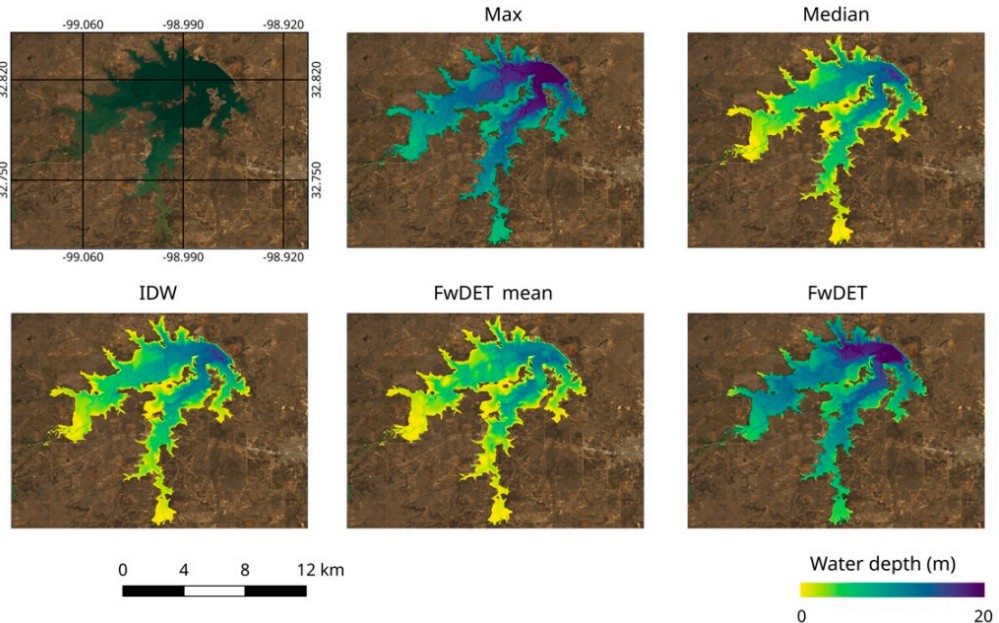

**Figure 12.** Hubbard Creek lake satellite image taken on 15 February 1991 and the different water depth maps obtained using the different methodologies.

### 3.4. Resolution and Slope

The decrease in the resolution (greater pixel size) leads to a decrease in the RMSE at very small pixel sizes. However, this decrease is greatest at a greater pixel resolution in lake Cawndilla (around 300 m; Figure 13a) compared to the Hubbard reservoir (50 m; Figure 13b). After the initial RMSE decrease, a steady increase in the RMSE is observed in both reservoirs, with a smaller increase for Hubbard lake. Although the overall RMSE is greater at Cawndilla, if the errors are normalized to the initial RMSE, the prediction error at Cawndilla lake is around two times the initial error, while at Hubbard Creek lake errors are about four times greater than the RMSE at the lowest pixel size. However, determination coefficients decrease faster in the Cawndilla lake, thus showing a faster deterioration of the precision of the volume prediction with pixel size.

The cumulative slope frequency distribution for the Cawndilla and Hubbard reservoirs is presented in Figure 13c. The slopes in the Cawndilla lake are much lower than the Hubbard reservoir, which has a much greater range of slopes. The design characteristics of both reservoirs indicate that the maximum depth of the Hubbard Creek reservoir is more than two times the depth of Cawndilla lake. This leads to the slightly different behaviors obtained when analyzing the resolution effect on the performance of volume estimations.

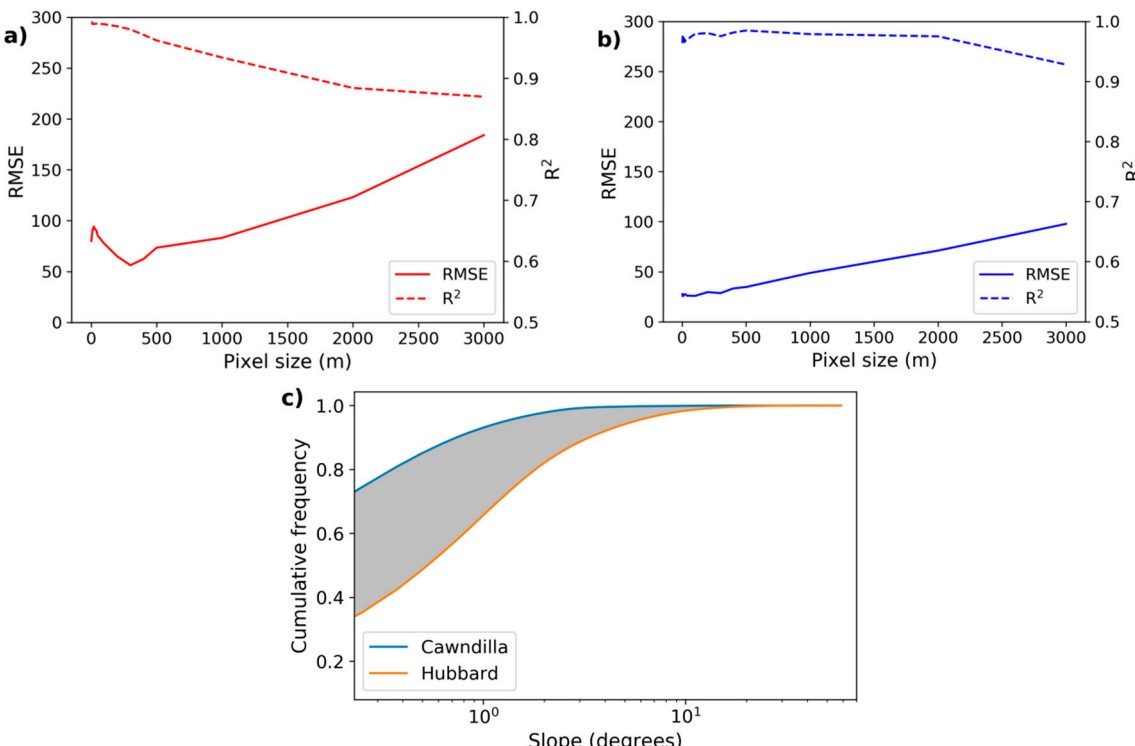

**Figure 13.** Effect of pixel size on the volume estimation errors for the Cawndilla (**a**) and the Hubbard (**b**) reservoirs. The slope differences between reservoirs are also presented (**c**).

## 4. Discussion

Several alternatives have been developed to deal with problems in estimating surface water volumes. Sawunyama et al. [37] found a power relationship between areas and volumes in small reservoirs within a catchment in South Africa, and Liebe et al. [38] estimated volumes of small reservoirs in Ghana based on their surface areas coupled with bathymetric data. While such estimates can be useful, extrapolation of the results to different reservoirs should be avoided because, as can be seen in this study, reservoirs have different area–volume relationships based on their particular bathymetries.

Regarding the pixel size effect on volume estimates, there have been several studies evaluating the horizontal resolution and its impact on hydrologic processes, with quite different results. Usery et al. [39], studying topographic indices, such as the topographic wetness index (TWI), which depends on the slope, concluded that between 3 and 30 m pixel sizes the results were unaffected, while a further reduction in resolution led to a degradation. In contrast, Sørensen and Seibert [40] found a considerable degradation in indices, moving from a 5 to a 10 m pixel size, but Cai and Wang [41] did not report any worsening by diminishing the resolution from 30 to 90 m. With respect to volumetric studies, Walczak et al. [42] showed that a decrease from 1 to 100 m resolution using LiDAR images led to approximately a 10% change in polder volumes. Additionally, the same study concluded that a 10% change in resolution led to little impact in the results. In the current study, the change of resolution in the flat areas of the terrain using the IDW methodology did not affect the volume estimates for a pixel size smaller than 300 m. There may even be a slight increase in performance as the resolution decreased from 1–3 m up to 300 m. Thus, despite the high DTM resolution, the accuracy of the methodology is constrained by the data used in the water detection, since there is a mismatch in the resolutions of the surface reflectance and the DTM data. For example, even if a 1 m resolution DTM with high vertical accuracy is used, the Landsat imagery used in the water detection constrains the water perimeter to a resolution of 30 m. This affects the volume estimates as elevations can suddenly change in an interval of 30 m, which is one of the reasons that might explain the increase in the performance when increasing the pixel size (Figure 13). However, this behavior may also be explained by the

flat surrounding terrain, and it could be different for steeper topographies surrounding lakes and reservoirs. Additionally, the 30 m resolution used when delineating the perimeter elevations causes considerably higher methodological errors in water volume estimates. In particular, this occurs when using methods that take extreme values in the distribution of elevations at the reservoir perimeters, such as the Max and FwDET methodologies, which lead to significant positive biases.

　　One of the difficulties analyzing the effect of the topography and the resolution of images on the volume estimates is that these cannot be taken into account in isolation because the pixel size affects the elevation on a pixel basis. In order to understand how slope affects area (and volume) estimates a diagram is presented (Figure 14). Constant slopes (Figure 14a) will offset gains (blue areas) and losses (red areas) in inundation area and volume at any pixel scale, independently of the slope angle. However for curved slopes, changes in the slope over distance (the second derivative of the elevation, or curvature; Figure 14b) cause biased errors in the volume estimates. In this last case, the increase in the pixel size and the higher curvature affect the errors in area and volume estimates by increasing the differences between the gains and losses in inundation area.

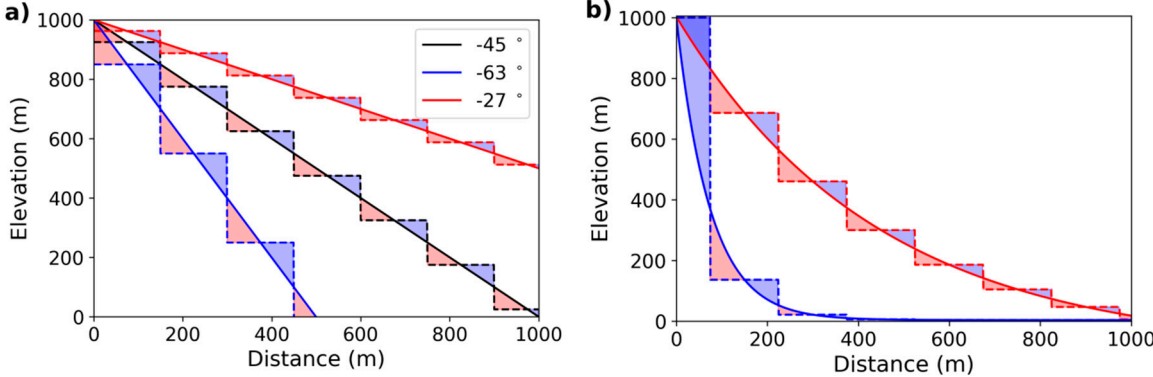

**Figure 14.** Area errors produced by pixel representation (dashed lines) of actual terrain elevation (continuous lines) for steady (**a**) and curved (**b**) slopes. Blue and red areas represent area gains and losses, respectively.

　　In this study, the curvature of the Cawndilla lake ranges between −0.22 and 0.23 with a standard deviation of 0.007, while the range and standard deviation of curvatures are significantly higher in the Hubbard Creek lake (from −1.13 to 1.21 and 0.05, respectively). This may explain the higher impact of increasing the pixel size on volume estimates at Hubbard Creek lake.

　　While the Median and the IDW methodologies are clearly superior, there is a clear and consistent negative bias in both. This systematic error may be a result of the digital terrain models, which were mostly acquired in the last 10 years, while the reference volumes (for example, calculated from original bathymetric surveys) date from 1987. In most cases, reservoir volumes reported are related to terrain conditions prior to the start of the operation of the reservoirs, a moment in which the terrain elevation was known. In lakes, these are generally linked to the date of the generation of a bathymetry, from which rating or elevation–area–volume curves are developed and usually associated with a gauge or limnimeter for continuous monitoring. However, sedimentation processes are continuously taking place at different rates, affecting the reservoir bathymetries, and this is generally not considered when monitoring the reservoir volumes [43], which implies an intrinsic error associated with the reference data. For example, the bathymetry used for the Hubbard Creek reservoir was carried out in 2018. It complemented a bathymetric survey carried out in 1997. The sedimentation processes between both surveys result in an estimated loss rate of storage capacity of 0.68 GL $y^{-1}$, which in 20 years accounts for a loss of 13.66 GL (Leber et al., 2018). In the case of Fork lake, it has sedimentation rates between 1.46 and 2.32 GL $y^{-1}$ [32], which results in 46.72 and 74.24 GL of lost capacity since the beginning of operations. However, these rates can hardly be applied continuously because they represent mean rates in the period of analysis, and sedimentation can abruptly change depending on rare climatic events.

In this study, since most of the bathymetries were surveyed in the last 10 years, a projection of the terrain to estimate past volumes must lead to lower estimates than the reported values (errors), due to lost reservoir storage capacity through sedimentation. However, the reported values based on limnimeters or gauges in the reservoirs are also over predicting the reservoir volumes because these are based on a projection of past conditions of the terrain. This partially explains the systematic negative errors obtained in the different reservoirs.

A way to deal with changing landscapes and bathymetries is to use terrain models obtained at higher frequency intervals. While such tasks may be unrealistic at a global scale, some alternatives that use remote sensing have been applied. For example, Zhang et al. [44], by using TanDEM-X imagery, were able to obtain bathymetric maps of several lakes in Brazil. Such an approach, if repeatedly carried out in time, can lead to a better knowledge of sedimentation rates and changes in landscape and water storage capacities.

Other sources of error, include the accuracy of the DTM sources. For instance, the LiDAR DTM used specifies a vertical accuracy of at least ±0.3 m and a horizontal accuracy of at least 0.8 m [28]. Even though no accuracies are reported in the bathymetric studies used, the USGS [45] reported vertical accuracies of 0.2, 0.28, and 0.46 m at the 95 percent confidence interval in the surveyed data, bathymetric model, and data contour map generated through echo sounder bathymetric studies, respectively. Therefore, even though the errors in high resolution DTM are low, these still may lead to large volume errors in big reservoirs.

Since terrain models and surface reflectance data at high spatial resolutions are progressively more available at global scales, these allow the monitoring of reservoirs and floods. While both the IDW and the Median methodologies had better performances than the rest, floods occurring in floodplain areas are usually taking place along river channels, which implies an elevation gradient. This precludes the use of techniques that assume a flat water surface, such as the Median methodology. Therefore, for such instances, the alternative IDW methodology might be a better choice for flood water volume estimates, which might be useful for water management plans and risk management studies. However, these estimates might have a stronger basis if the temporal resolution and the operation times of the datasets allow for continuous and long-term monitoring, which might also allow analyzing the temporal dynamic of such events [46].

While other approaches to study surface water dynamics have been developed by coupling satellite imagery and altimetry data [47,48], these methods only provide data on fluctuations in volumes and not total volumes, which constrains their applicability for water management planning.

## 5. Conclusions

The Max and the FwDET methodologies resulted in considerable errors and high bias in volume estimation. Both methods were strongly affected by a combination of issues, including: mixed land covers within surface reflectance image pixels, a mismatch between DTM and surface reflectance product resolutions, and errors in the water detection step. The FwDET_mean method had an intermediate performance. Both the Median and IDW methodologies outperformed the rest across the studied reservoirs. However, a negative bias was systematically observed in the estimates.

Pixel size and the curvature of the terrain were common factors introducing errors that affect area and volume estimations. Other sources of systematic errors are associated with the terrain models and the reported volumes stored in reservoirs. These last fail to consider the bathymetric changes occurring in reservoirs due to sedimentation processes, which can lead to an overestimation of water availability.

Even though some relationships between reservoir areas and volumes have been discussed in the literature, the extrapolation of such results to different scenarios must be avoided since area–volume relationships are specific to each reservoir and depend on their specific bathymetries.

Future research using higher resolution imagery or processed datasets for water detection, including synthetic aperture radar, could be used to improve the results. Moreover, the Surface Water Ocean Topography mission, projected for 2021, will allow a more thorough understanding of

surface waters, but different alternatives will still be required to study the dynamics of hydrological processes and their recurrence since decades of available data will still be required to study long term hydrologic processes.

**Author Contributions:** Original idea, I.F.; coding, I.F.; supervision, R.W.V.; draft writing, I.F.; draft edition, J.P., F.v.O., R.W.V. and I.F.

**Funding:** This research received no external funding.

**Acknowledgments:** The first author was supported through a scholarship by the Chilean Government ("Becas Chile", Comisión Nacional de Investigación Científica y Tecnológica, CONICYT).

**Conflicts of Interest:** The authors declare no conflict of interest.

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
