# Peer review of "Comparison of Surface Water Volume Estimation Methodologies That Couple Surface Reflectance Data and Digital Terrain Models"

_water, doi:10.3390/w11040780_

Round 1

Reviewer 1 Report

The manuscript presents a new water depth calculation methodology which uses remote sensing water extent layer and a GIS terrain-based (IDW interpolation) approach. The methodology is tested in multiple reservoirs by comparing the calculated and measured water storage volumes. It is also compared against two similar water depth calculation approaches. The paper is well written and the methodology and the evaluation approach are robust. There is one major issue that needs to be corrected: the way the authors applied FwDET is most likely wrong. FwDET's Focal Statistics iterations extract the maximum value, not the mean as is described and likely applied in this study. It is also implied in the paper that the low-pass filter is applied at every iteration which is not the case - it is only applied at the end of the script to produce a 2nd output raster (the tool also output the original water depth raster). I am attaching FwDET Python script (which is freely available online). Both of these potential errors may explain (as the authors indeed assert) FwDET's under-predictions. In actuality, the methodology the authors use may have improved FwDET predictions for these case studies by smoothing it (which may be reasonable for large lakes). If indeed the authors did not accurately execute FwDET, as oppose to not properly describing it, I suggest that they redo the analysis. Keeping the existing results may be interesting as well (so presenting both the original and modified FwDET). 

The description of the new IDW methodology needs to be expended. I will provide specific points below but generally, given its applicability, the description should be detailed enough to allow others to repeat it. I also suggest that the author add a figure that shows, side by side, the water depth maps produced by the three methods, at least for selected case study(s). 

I enjoyed reading this paper and think it should be published once these issues are addressed.

Good luck

Specific comments:

Line 20 and later - change 'Cohen' to 'FwDET'.

Line 75-76 - flooding analysis is not discussed in the manuscript. I suggest you add a discussion about the potential of the new approach or remove this from the introduction.

Lines 113-115 - this may not be clear to most people. Explain that (I assume) the data you had was water depth, not bathymetry.

Table 1 and later - the use of gigaliters is less intuitive than m3 or km3 in my view. Maybe use both (GL and km3).

Line 139 - this sounds like you are manually digitizing the water-extent from each image - explain that these are end-members.   

Line 160 - indicate that the tool was developed for flood analysis.

Line 162 and later - as explained above - use maximum value for the focal statistics neighborhood. 

Line 168 - as explained above - additional iterations will not further smooth the results but enough iterations are indeed needed to cover the water extent domain.

Line 172 - an example of an insufficient description of the methodology - what do you mean by 'sampling'? Do you randomly (or at some interval) select DEM cells along the perimeter or are you using all of them?  If the former, how many or at what interval? if the latter, explain that better. Provide some details about the software and tools used.

Line 177-178 - Did you get such small polygons in these large lakes? If so under what circumstances?

Figure 8 is not very useful in my view.

Line 229 - add a comma after 'analyses' (should be analysis?)

End of section 3.2 - what does that mean for future use of this approach? this will be quite a common situation - what do you propose one do (maybe nothing)? Expand!

Lines 275-277 - why? discuss!

Lines 289 and 290 - should it be 'increase' in RMSE? 

First paragraph in the Discussion - I think this is an important point. May have important applicable potential. I suggest you mention it in the conclusions. 

Paragraph starting in line 372 - doesn't it contradicts your last argument?

Last paragraph - you should further discuss (likely in the results or discussion sections) issues with the remote sensing products and how spatial mismatch with the DEM can lead to errors. 

Author Response

Thanks for the comments and suggestions, these have improved the manuscript.

comment 1: "There is one major issue that needs to be corrected: the way the authors applied FwDET is most likely wrong. FwDET's Focal Statistics iterations extract the maximum value, not the mean as is described and likely applied in this study. It is also implied in the paper that the low-pass filter is applied at every iteration which is not the case - it is only applied at the end of the script to produce a 2nd output raster (the tool also output the original water depth raster). I am attaching FwDET Python script (which is freely available online). Both of these potential errors may explain (as the authors indeed assert) FwDET's under-predictions. In actuality, the methodology the authors use may have improved FwDET predictions for these case studies by smoothing it (which may be reasonable for large lakes). If indeed the authors did not accurately execute FwDET, as oppose to not properly describing it, I suggest that they redo the analysis. Keeping the existing results may be interesting as well (so presenting both the original and modified FwDET)."

Response to comment 1: We modified the methodology based on the reviewer’s suggestions and we also left the change that we implemented. Since we just used the methodology based on the article, it wasn't clear what kind of focal statistic the authors used (the authors mostly refer to "focal statistics" in the text). Therefore, we modified the function based on the attached script that you sent. The low-pass filter was indeed applied after the iterations. We modified the paragraph in the methodology:

"The third method (FwDET) was developed by Cohen et al. [17] for flood analysis. It involves the conversion of inundated areas into polygons to obtain the elevations at the perimeter of polygons. Subsequently, it applies a focal statistic (focal maximum) in a series of iterations with increasing windows size to populate the area inside the polygons with water elevations. The final stage involves subtracting the water elevations from the original DTM to get the water depths. Negative water depths are converted to 0, and a final low-pass filter with a kernel of 3 pixels is used to smooth any abrupt change in the water elevations. An important detail is that the number of iterations corresponds to the minimum number of iterations needed to completely populate all the inundation polygons. Additionally, a modification of the FwDET methodology was implemented (subsequently referred to as FwDET_mean), which replaced the focal maximum statistic of the original methodology with a focal mean statistic."

comment 2: The description of the new IDW methodology needs to be expended. I will provide specific points below but generally, given its applicability, the description should be detailed enough to allow others to repeat it. I also suggest that the author add a figure that shows, side by side, the water depth maps produced by the three methods, at least for selected case study(s).

Response to comment 2: We expanded the details of the methodology used considering this and subsequent minor comments. We also added an image with a specific date showing the differences in the methods as requested.

comment 3: Line 20 and later - change 'Cohen' to 'FwDET'.

Response to comment 3: The change was accepted.

comment 4: Line 75-76 - flooding analysis is not discussed in the manuscript. I suggest you add a discussion about the potential of the new approach or remove this from the introduction.

Response to comment 4:  A paragraph regarding floods was included in the discussion: “Since terrain models and surface reflectance data at high spatial resolutions are progressively more available at global scales, these allow the monitoring of reservoirs and floods. While both, the IDW and the Median methodologies had better performance than the rest, floods occurring in floodplain areas are usually taking place along river channels, which implies an elevation gradient. This precludes the use of techniques that assume a flat water surface, such as the Median methodology. Therefore, for such instances, the alternative IDW methodology might be a better choice for flood water volume estimates, which might be useful for water management plans and risk management studies. However, these estimates might have a stronger basis if the temporal resolution and the operation times of the datasets allow for a continuous and long-term monitoring, which might also allow analysing the temporal dynamic of such events [46]. ”    

comment 5: Lines 113-115 - this may not be clear to most people. Explain that (I assume) the data you had was water depth, not bathymetry.

Response to comment 5: No, actually we did use bathymetric data to estimate the water depths as the document explains. This was further addressed by specifying that water depths were obtained from DTM products, which include bathymetries.

comment 6: Table 1 and later - the use of gigaliters is less intuitive than m3 or km3 in my view. Maybe use both (GL and km3).

Response to comment 6: All tables were changed according to your suggestion and in the first image were the y-axis points out to GL, it was included in the description that 1 GL = 10⁶ m³.

comment 7: Line 139 - this sounds like you are manually digitizing the water-extent from each image - explain that these are end-members.   

Response to comment 7: Suggestion accepted. The paragraph now reads "Therefore, a Classification and Regression Tree analysis (CART) was applied to the reflectance bands of the Landsat imagery to delineate inundated areas by selecting known surface water and dry land end-members. This was done by drawing polygons classified as water on several images, previously masked to remove clouds and cloud shadows. Other polygons were also delineated on dry areas with different land covers, and classified as dry polygons, which produced surface water classified images (Figure 7)."

comment 8: Line 160 - indicate that the tool was developed for flood analysis.

Response to comment 8: Suggestion accepted.

comment 9: Line 162 and later - as explained above - use maximum value for the focal statistics neighborhood.

Response to comment 9: Suggestion accepted, see response to comment 1.

comment 10: Line 168 - as explained above - additional iterations will not further smooth the results but enough iterations are indeed needed to cover the water extent domain.

Response to comment 10: We deleted the sentence that pointed out that a further increase in the iterations would lead to lower volume estimates.

However, we do disagree somewhat, since it will depend on the focal statistic used. For example, using the Focal maximum the estimates should increase up to being equal to the volumes obtained assuming the maximum elevation in the perimeter of inundated areas as the water elevation. This is because the increase in iterations increases the propagation of the maximum elevation in the inundated areas. In the case of the Focal mean statistic, the water elevations should decrease with the increase in iterations until being equal to using the mean elevation in the perimeter as the water elevation due to the propagation of the mean elevation.

comment 11: Line 172 - an example of an insufficient description of the methodology - what do you mean by 'sampling'? Do you randomly (or at some interval) select DEM cells along the perimeter or are you using all of them?  If the former, how many or at what interval? if the latter, explain that better. Provide some details about the software and tools used.

Response to comment 11: We adjusted the method with your suggestions and this now reads:

"The new methodology consists of delineating the perimeter of inundated areas and applies a random sampling of the perimeter elevations using a buffer of 2.5 m each side of the perimeter contour, setting the number of sampling points to 5000. Then, an inverse distance weighting interpolation is applied to the sampled points to obtain the elevation of the water, which is subsequently subtracted from the DTM to obtain the water depths (Figure 8). ... All preprocessing steps and the different methodologies were implemented in Google Earth Engine."

comment 12: Line 177-178 - Did you get such small polygons in these large lakes? If so under what circumstances?

Response to comment 12: Sure, in some cases we could identify some small reservoirs surrounding the lakes, which had a high recurrence in time. In the image attached a map of the recurrence of water is shown in the Tawakoni lake, the scale goes from 0: yellow, to 1: blue. Surrounding the lake, several small reservoirs can be seen which have a high recurrence (they are there almost all the time).

The same can be observed for example in the Ray Roberts lake:

comment 13: Figure 8 is not very useful in my view.

Response to comment 13: We believe that it graphically demonstrates what we explained in the methodology, which was criticized as incomplete.

comment 14: Line 229 - add a comma after 'analyses' (should be analysis?)

Response to comment 14: suggestion accepted.

comment 15: End of section 3.2 - what does that mean for future use of this approach? this will be quite a common situation - what do you propose one do (maybe nothing)? Expand!

Response to comment 15: We weren’t totally sure of what part of the document you were referring to. We assumed that you refer to the removal of the bias in the estimates due to the use of a reference elevation in the bathymetric survey.

We added some potential solutions to this problem: “One solution to this problem might be to use the mean elevation of the surveyed days rather than the mean elevation of the reservoir for the generation of the bathymetry maps, especially if the variation of water depths during the survey is lower than the variation in the entire reservoir monitoring period.  

Additionally, this bias might be simply removed from the estimates assuming that it is caused by using the mean level elevation of the reservoir instead of the water level at the moment of the survey.”

comment 16: Lines 275-277 - why? discuss!

Response to comment 16: An explanation was included: “A greater reservoir storage volume relates to a greater flooded area. Since the perimeter of reservoirs in surface reflectance images is often composed of mixed land surfaces (flooded and dry land), these areas are more susceptible to water detection classification errors. These errors and the mismatch between the resolution of DTMs and the Landsat images can cause errors in the elevations at the perimeter of the reservoirs, used to fill the water elevations. This results in bigger errors of volume estimates associated with greater inundated areas. ”    

comment 17: Lines 289 and 290 - should it be 'increase' in RMSE?

Response to comment 17: Actually is a decrease in the RMSE which is observed in Figure 12. This is later mentioned and discussed in the next section.

comment 18: First paragraph in the Discussion - I think this is an important point. May have important applicable potential. I suggest you mention it in the conclusions.

Response to comment 18: A small paragraph was added to the conclusions regarding this point.

comment 19: Paragraph starting in line 372 - doesn't it contradicts your last argument?

Response to comment 19: The previous paragraph highlights that sedimentation processes have led to a reduction in storage capacity, and these rates aren't continuous. The next paragraph continues analysing sedimentation as a source of errors that should cause: a positive bias in the reported volumes since sedimentation rates are not being considered (it is estimated that there is more water than the actual volumes); but also a negative bias in modeled volumes when recent terrain elevation data is extrapolated to past conditions, since the reservoir capacity was greater. We don't think that there is a contradiction.

comment 20: Last paragraph - you should further discuss (likely in the results or discussion sections) issues with the remote sensing products and how spatial mismatch with the DEM can lead to errors.

Response to comment 20: Please see response to comment 16. Additionally, a paragraph discussing the DTM sources accuracies was included in the discussion section: “Other sources of error, include the accuracy of the DTM sources. For instance, the LiDAR DTM used specifies a vertical accuracy of at least ±0.3 m and a horizontal accuracy of at least 0.8 m [28]. Even though no accuracies are reported in the bathymetric studies used, the USGS [45] reported vertical accuracies of 0.2, 0.28 and 0.46 m at the 95 percent confidence interval in the surveyed data, bathymetric model and data contour map generated through echo sounder bathymetric studies, respectively. Therefore, even though the errors in high resolution DTM are low, these still may lead to large volume errors in big reservoirs.“

Reviewer 2 Report

The research investigates surface water volume estimation which is an important topic in Environmental Management. However, the methodologies developed or used might not be appropriate for this application and oversimplified the problem.

First, according to the title, abstract and conclusion, surface reflectance data (from Landsat imagery) has been used for detecting the inundated extent in 2D. However, the supervised classification has not been explained in the paper at all. The classification method used for the water surface detection should be explained (in detail) as the water volume estimation highly depends on the accuracy of the water surface (polygon – the classification results) as well as topography of the study areas. The accuracy of water areas depends on the classification method, the number of training samples, etc.

Have you programmed the classification method or you used a software (in this case –what software)?

The concept of Max methodology is not accurate (as the results confirmed) and I am wondering why the authors considered it for comparison. The concept of hydroflattening - that assumes the water surface is locally flat – seems logic and many software and efficient algorithms use it. However, regarding noises in DEM, using the maximum elevation of inundated area to estimate the water surface is not correct. A noise removal preprocessing should be applied or median surface should be considered for the water surface estimation.

In the introduction section, the temporal dynamics of water surface is discussed. It’d be interesting to include a time series analysis in the paper to show the performance of your method (e.g. water surface extraction over one of the study areas in different times).

Are DEM (digital elevation model) and DTM (digital terrain model) used for same concept - bare earth elevation-  in the paper? To keep consistency, one of them should be used.

The accuracy analysis for water volume estimation is not enough. In addition to the data resolution, the accuracy of the dataset itself affects the accuracy of the results that should be considered (including imagery, classification method, DEM/DTM, USGS data, interpolation/sampling, cloud filtering, etc.).

Author Response

Thanks for the comments and suggestions, these have improved the manuscript.

comment 1: First, according to the title, abstract and conclusion, surface reflectance data (from Landsat imagery) has been used for detecting the inundated extent in 2D. However, the supervised classification has not been explained in the paper at all. The classification method used for the water surface detection should be explained (in detail) as the water volume estimation highly depends on the accuracy of the water surface (polygon – the classification results) as well as topography of the study areas. The accuracy of water areas depends on the classification method, the number of training samples, etc.

Response to comment 1: We used 188 polygons in areas where surface water was present and 205 polygons of dry land areas for training the classifier. However, Google Earth Engine uses all the pixels in these polygons rather than a statistical reduction of the data by polygon. A paragraph describing these details in the classification was added to the methods.

Additionally, we tested the classification using three other Landsat images over USA and Australia, and we delineated 30 polygons with known surface water and 30 polygons in dry land areas. Then, we obtained the overall accuracy and the kappa coefficient of the classification. The results gave 0.99 and 0.99 for both, overall accuracy and kappa coefficient using the training dataset. The validation dataset leads to similar results, but with a very slight decrease in both coefficients (see attached picture). This doesn’t necessarily mean that the classification is perfect, but it matches the visual classification that was carried out when selecting the polygons.

comment 2: Have you programmed the classification method or you used a software (in this case –what software)?

Response to comment 2: The classification was carried out using google earth engine, so it was scripted in Java.

comment 3: The concept of Max methodology is not accurate (as the results confirmed) and I am wondering why the authors considered it for comparison. The concept of hydroflattening - that assumes the water surface is locally flat – seems logic and many software and efficient algorithms use it. However, regarding noises in DEM, using the maximum elevation of inundated area to estimate the water surface is not correct. A noise removal preprocessing should be applied or median surface should be considered for the water surface estimation.

Response to comment 3: We agree that it’s not accurate. However, in theory the maximum elevation inundated in a reservoir should correspond to the water level if the surface water was flat. In addition, this method has empirically been used in the study cited. However, since we agree with your criticism of the methodology we added the median elevation in the perimeter of the reservoirs to the methods used, which certainly improved the results.

comment 4: In the introduction section, the temporal dynamics of water surface is discussed. It’d be interesting to include a time series analysis in the paper to show the performance of your method (e.g. water surface extraction over one of the study areas in different times).

Response to comment 4: We didn’t want to add more images because the document already has 13. With the revision, and due to the request of another reviewer we added another image, and that makes 14. We think that the number of figures is already enough and that a plot with the time series of one of the reservoirs doesn’t do much difference, and therefore, we decided not to include more figures. However, here you can see, for example, the Stanley Draper lake with the time series of reported volumes and the predicted volumes using three of the different methodologies.

comment 5: Are DEM (digital elevation model) and DTM (digital terrain model) used for same concept - bare earth elevation-  in the paper? To keep consistency, one of them should be used.

Response to comment 5: Suggestion accepted.

comment 6: The accuracy analysis for water volume estimation is not enough. In addition to the data resolution, the accuracy of the dataset itself affects the accuracy of the results that should be considered (including imagery, classification method, DEM/DTM, USGS data, interpolation/sampling, cloud filtering, etc.).

Response to comment 6: Two additional paragraphs were added commenting on the errors from the DTM sources and the mismatch of resolutions between the DTM and the surface reflectance data:

“A greater reservoir storage volume relates to a greater flooded area. Since the perimeter of reservoirs in surface reflectance images is often composed of mixed land surfaces (flooded and dry land), these areas are more susceptible to water detection classification errors. These errors and the mismatch between the resolution of DTMs and the Landsat images can cause errors in the elevations at the perimeter of the reservoirs, used to fill the water elevations. This results in bigger errors of volume estimates associated with greater inundated areas.

… Other sources of error, include the accuracy of the DTM sources. For instance the LiDAR DTM used specifies a vertical accuracy of at least ±0.3 m and a horizontal accuracy of at least 0.8 m [28]. Even though no accuracies are reported in the bathymetric studies used, the USGS [45] reported vertical accuracies of 0.2, 0.28 and 0.46 m at the 95 percent confidence interval in the surveyed data, bathymetric model and data contour map generated through echo sounder bathymetric studies, respectively. Therefore, even though the errors in high resolution DTM are low, these still may lead to large volume errors in big reservoirs.”

Reviewer 3 Report

The research presented here is quite straightforward where lake volumes across various regions of the world are calculated using remote sensing. Digital terrain models are used in conjunction with three methods to compute water elevation and then laek volume.

The paper is well written and easy to follow. The paper itself does not contribute towards a novel idea but provides an application of existing technology and highlights several pertinent issues in lake volume computation.

Major comment

As the authors point out in the discussion, there appears to be significant negative bias in volume estimation. This seems to be the major issue in this study and the authors address as potentially being introduced as a result of sedimentation. It also appears that there is an area/volume relationship – larger lakes have a higher residual. Since negative bias is an important aspect of this study, I ask the authors to add further analyses clarifying the cause separating these factors further. In line 170, the authors also note that the Cohen et al (2017) methodology underestimated volumes so whether clarification might be required as to explain this bias is methodological or as a result of natural processes (e.g. sedimentation) as the authors here claim.

Minor comments

Line 46. By quantity, you mean volume but perhaps better to clarify right on this sentence.

Line 140. For CART, were normalized indices used as inputs (e.g. NDWI). This would potentially improve the delineation of inundated water bodies. This classification probably needs to be discussed more in detail here. At present, it just glosses over this step.

Figure 10. In this instance, larger lake volumes appear to be consistently underestimated. The lake area, volume relation and the negative bias needs to be clarified a bit further.

Author Response

Thanks for the comments and suggestions, these have improved the manuscript.

Major comment 1: As the authors point out in the discussion, there appears to be significant negative bias in volume estimation. This seems to be the major issue in this study and the authors address as potentially being introduced as a result of sedimentation. It also appears that there is an area/volume relationship – larger lakes have a higher residual. Since negative bias is an important aspect of this study, I ask the authors to add further analyses clarifying the cause separating these factors further. In line 170, the authors also note that the Cohen et al (2017) methodology underestimated volumes so whether clarification might be required as to explain this bias is methodological or as a result of natural processes (e.g. sedimentation) as the authors here claim.

Response to Major comment 1: The direct relationship between the residuals and the reservoir volume estimates was discussed in the following paragraph: “A greater reservoir storage volume relates to a greater flooded area. Since the perimeter of reservoirs in surface reflectance images is often composed of mixed land surfaces (flooded and dry land), these areas are more susceptible to water detection classification errors. These errors and the mismatch between the resolution of DTMs and the Landsat images can cause errors in the elevations at the perimeter of the reservoirs, used to fill the water elevations. This results in bigger errors of volume estimates associated with greater inundated areas.”

We also tried to address the methodological reason for the errors using the Cohen methodology (note that we changed Cohen by FwDET since other reviewer requested this): “The FwDET methodology leads to high R2 coefficients, but also to high errors in the Menindee and Pamamaroo lakes. In this case, the bias is positive due to a propagation of the maximum elevations in the successive iterations of the focal statistic, and therefore propagates the errors caused in the water detection and by the mismatch between the resolutions of the DTM and the surface reflectance product.”

We also added a paragraph pointing out the accuracy of the DTM sources used: “Other sources of error, include the accuracy of the DTM sources. For instance the LiDAR DTM used specifies a vertical accuracy of at least ±0.3 m and a horizontal accuracy of at least 0.8 m [28]. Even though no accuracies are reported in the bathymetric studies used, the USGS [45] reported vertical accuracies of 0.2, 0.28 and 0.46 m at the 95 percent confidence interval in the surveyed data, bathymetric model and data contour map generated through echo sounder bathymetric studies, respectively. Therefore, even though the errors in high resolution DTM are low, these still may lead to large volume errors in big reservoirs.”

We agreed that it would be fantastic to be able to separate the causes of errors and quantify them in isolation. However, such a task seems impossible to do with the available data that we have. We tried for example to evaluate the water detection step alone by estimating the overall accuracy and the kappa coefficient of the classification. The results gave 0.99 and 0.99 for both, overall accuracy and kappa coefficient using training and validation datasets. When quantifying the errors of the methodologies, we tried to explain sources that could affect these. However, quantifying them separately would even with a lot of monitoring devices be very difficult. We tried to give some examples of volumetric errors caused by sedimentation, we also added the errors inherent in the DTM data, and we explained that there are errors associated to the reference data, which makes even harder to get an estimation of errors. We hope that with the modifications carried out the whole picture associated with errors might be clarified a little bit further.  

Minor comment 1: Line 46. By quantity, you mean volume but perhaps better to clarify right on this sentence.

Response to Minor comment 1: yes, we meant volume. We clarified this further in the sentence.

Minor comment 2: Line 140. For CART, were normalized indices used as inputs (e.g. NDWI). This would potentially improve the delineation of inundated water bodies. This classification probably needs to be discussed more in detail here. At present, it just glosses over this step.

Response to  Minor comment 2: We didn’t try normalised difference indices (NDVI, NDWI, modified NDWI, ...) because we wanted to include all surface reflectance bands in the water detection, since we cited a study that concludes that using a supervised classification over Landsat images gives better results. However, as suggested, we expanded the description of the water classification.

Minor comment 3: Figure 10. In this instance, larger lake volumes appear to be consistently underestimated. The lake area, volume relation and the negative bias needs to be clarified a bit further.

Response to Minor comment 3: See Response to Major comment 1.

Round 2

Reviewer 2 Report

The paper has been improved. 

Water EISSN 2073-4441 Published by MDPI AG, Basel, Switzerland RSS E-Mail Table of Contents Alert
Back to Top